# Influence of Pulse Duration on X-ray Emission during Industrial Ultrafast Laser Processing

**DOI:** 10.3390/ma15062257

**Published:** 2022-03-18

**Authors:** Julian Holland, Rudolf Weber, Marc Sailer, Thomas Graf

**Affiliations:** 1Institut für Strahlwerkzeuge, Universität Stuttgart, Pfaffenwaldring 43, 70569 Stuttgart, Germany; rudolf.weber@ifsw.uni-stuttgart.de (R.W.); thomas.graf@ifsw.uni-stuttgart.de (T.G.); 2TRUMPF Laser GmbH, Aichhalder Straße 39, 78713 Schramberg, Germany; marc.sailer@trumpf.com

**Keywords:** laser plasma, X-ray emission, dose rates, pulse duration dependence, ultrafast laser processing, hot-electron temperature

## Abstract

Soft X-ray emissions during the processing of industrial materials with ultrafast lasers are of major interest, especially against the background of legal regulations. Potentially hazardous soft X-rays, with photon energies of >5 keV, originate from the fraction of hot electrons in plasma, the temperature of which depends on laser irradiance. The interaction of a laser with the plasma intensifies with growing plasma expansion during the laser pulse, and the fraction of hot electrons is therefore enhanced with increasing pulse duration. Hence, pulse duration is one of the dominant laser parameters that determines the soft X-ray emission. An existing analytical model, in which the fraction of hot electrons was treated as a constant, was therefore extended to include the influence of the duration of laser pulses on the fraction of hot electrons in the generated plasma. This extended model was validated with measurements of H (0.07) dose rates as a function of the pulse duration for a constant irradiance of about 3.5 × 10^14^ W/cm^2^, a laser wavelength of 800 nm, and a pulse repetition rate of 1 kHz, as well as for varying irradiance at the laser wavelength of 1030 nm and pulse repetition rates of 50 kHz and 200 kHz. The experimental data clearly verified the predictions of the model and confirmed that significantly decreased dose rates are generated with a decreasing pulse duration when the irradiance is kept constant.

## 1. Introduction

Processing laser material using ultrafast lasers with a pulse durations of less than about 10 ps allows for very precise processing of a wide range of materials due to the significantly reduced thermal interaction between the laser beam and the remaining bulk material [1,2,3]. In recent years, both the average power and the pulse energy of ultrafast lasers have been continuously increased to more than 10 kW [4,5,6] and more than 200 mJ [7], allowing the forecast of a continuous steady increase [8]. This promises a significant increase in productivity within the next few years. Today, industrially available lasers provide pulse energies of up to several 100 µJ, at average powers of up to a few hundred Watts. Irradiances of up to several 10^15^ W/cm² can be achieved when the laser beam is tightly focused on a plane surface. Such high irradiances lead to strong plasma generation resulting in the emission of bremsstrahlung, recombination radiation, and line emission, with photon energies exceeding 5 keV and even up to tens of keV [9]. This X-ray emission might lead to possibly hazardous dose rates for an operator. Today, it is generally presumed to be a safety issue that resulted in new regulations and numerous investigations [10,11,12,13,14,15,16].

During the interaction of a laser pulse with material, free electrons are created by the leading edge of the pulse [17]. The free electrons are further excited by the remainder of the pulse due to different mechanisms, such as inverse bremsstrahlung and resonance absorption [18], resulting in a fraction of electrons with high kinetic energy and temperatures of several keV [9], which are commonly referred to as hot electrons [17]. This fraction of hot electrons is responsible for the X-ray emission considered in this paper. The expansion of plasma during the laser pulse determines the efficiency of the energy transfer from laser radiation to the electrons of the plasma and, therefore, influences the fraction of hot electrons. How far the plasma expands during the laser pulse is calculated by the expansion velocity times the pulse duration. This means that the pulse duration is an important parameter with respect to the amount of X-radiation emitted from processes that are performed with ultrashort laser pulses.

The model described in [19] allows for the calculation of X-ray emissions and the resulting dose rate during ultrafast laser processing. However, in this model, the fraction of hot electrons was treated as an arbitrary constant used to calibrate the model, with experimental values for the dose rate resulting from processing with a pulse duration of 1 ps. Therefore, for this paper, the model was modified to include the dependence of the number density of hot electrons on the pulse duration. This modified model was calibrated with measurements of the dose rates resulting from laser processing of stainless steel (1.4301). The experiments were performed at three different processing facilities using laser sources with irradiances in the range from 5.4 × 10^12^ W/cm^2^ up to 3.6 × 10^14^ W/cm^2^ and durations of laser pulses ranging between 75 fs and 5 ps.

## 2. Influence of the Pulse Duration on the Dose Rate

A semi-heuristic modification of the analytical model presented in [19] is derived below to include the experimentally observed influence of pulse duration on dose rate.

### 2.1. Spectral Power of Emission

The energy distribution of the hot electrons is responsible for the emission of the soft X-rays with photon energies >5 keV [17] that are of interest in this paper. The spectral power of the bremsstrahlung emitted by these hot electrons can be approximated by [17,19]
(1)$$\frac{dP_{B}}{d\omega }\;=\;V_{P}\cdot \frac{32\pi }{3}\cdot \sqrt{\frac{2\pi }{3m_{e}\cdot k_{B}\cdot T_{h}}}\cdot \frac{Z_{i}\cdot e^{6}\cdot n_{h}^{2}}{m_{e}\cdot c^{3}}\cdot e^{-\frac{\hbar \cdot \omega }{k_{B}\cdot T_{h}}},$$
where ℏ and *k_B_* are the Planck constant divided by 2π and the Boltzmann constant, respectively; c is the speed of light; and *m_e_* and *e* are the mass and charge of the electron, respectively. The properties of the plasma that are relevant for the X-ray emission are its volume *V_P_*, the electron temperature *k_B_T_e_*, (usually expressed as an energy and given in eV), the degree of ionization *Z_i_*, and the number density *n_h_* of the hot electrons. The exponential term in Equation (1) describes the spectral distribution that is solely determined by the temperature of the hot electrons. The determination of *V_P_* and *Z_i_* in the model is extensively discussed in [19] and not repeated here.

The scaling of *T_h_* as a function of the wavelength-corrected laser irradiance $\lambda_{L}^{2}I_{0}$ was experimentally determined for the industrial processing conditions [19] considered here for irradiances ranging from *I*_0_ ≈ 10^12^ W/cm^2^ to *I*_0_ ≈ 10^15^ W/cm^2^, yielding [9,20,21,22,23,24]
(2)$$T_{h}\;=\;c_{T}\cdot {\left(\lambda_{L}^{2}\cdot I_{0}\right)}^{S},$$
for *T_h_* in keV, with *c_T_* = 4.1 × 10^−8^ keV and *s* = 0.53, *λ_L_* in µm, and *I*_0_ in W/cm^2^ given by
(3)$$I_{0}\;=\;\frac{2E_{P}}{t_{P}\cdot \pi \cdot r_{F}^{2}}$$
where *E_P_* is the energy of the laser pulse, *t_P_* is the pulse duration, and *r_F_* is the radius of the laser beam on the surface of the irradiated workpiece.

### 2.2. Number Density of Hot Electrons

In [19], the number density *n_e_* of the total amount of free electrons was calculated from the number density of the ions assuming quasi-neutrality and an average degree of ionization of *Z_i_*. Only a small fraction *q_h_* of the total amount of electrons is excessively heated to a hot-electron temperature *T_h_*. These hot electrons are created when *n_e_* is close to the critical density [17], where the plasma frequency equals the laser frequency, by various mechanisms, including the important effect of resonance absorption [18]. In [19], the fraction *q_h_* was supposed to be constant, resulting in the number density of hot electrons given by
(4)$$n_{h}\;=\;q_{h}\cdot n_{e}\;=\;q_{h}\cdot Z_{i}\cdot n_{i}$$

The fraction *q_h_* was used as the sole fit parameter for the model. In the case of laser pulses with a duration of 1 ps, a wavelength of 1 µm, and processing of steel, *q_h_* was found to amount to 5.5 × 10^−4^.

### 2.3. Modified Fraction of Hot Electrons

The expansion of a laser-produced plasma can be considered as purely perpendicular to the surface (the z-direction discussed below) in the initial stage [25,26], which is considered here. The density gradient of the electrons *∂n_e_*/*dz* at the location of the critical electron density is of particular importance for the fraction of hot electrons, which is created during the laser pulse ([17,25]); shallower gradients enhance the laser–plasma interaction and, with it, the fraction of hot electrons. During the laser pulse, the initial extent of the plasma in the z-direction, $\ell_{i}$, which is in the range of the optical penetration depth of tens of nanometers in metals, increases with the ion’s speed of sound [25,26], i.e.,
(5)$$\ell_{P}\left(t\right)\;=\;\ell_{i}\;+\;c_{S}\cdot t,$$
where the speed of sound for ions of the mass *m_i_* is given by
(6)$$c_{S}=\sqrt{\frac{Z_{i}\cdot k_{b}\cdot T_{h}}{m_{i}}}.$$

This means that *∂n_e_*/*dz* at the location of the critical electron density decreases during the laser pulse. A linear density gradient can be assumed during the first phase of expansion of the plasma [25]. For the sake of simplicity, the gradient *∂n_e_*/*dz* at the end of the laser pulse, where $c_{S}t\;\gg \;\ell_{i}$ also holds, was used to describe the dependence of the X-ray emission on the pulse duration. In this simplified case, *∂n_e_*/*dz* is proportional to $\ell_{P}\left(t_{P}\right)$, allowing for the consideration discussed below.

According to Equations (2) and (3), the resulting temperature T_h_ of the hot electrons at the end of the pulse approximately scales with $T_{h}\propto t_{P}^{-1/2},$ and according to Equation (6), *c*_S_ is proportional to *T_h_*^1/2^. Considering that these dependencies lead to the conclusion that c_s_ approximately scales with *t_P_*^−1/4^, and, therefore, $\ell_{P}\left(t_{P}\right)$ is found to be proportional to *t_P_*^3/4^.

As the model in [19] was calibrated with a pulse duration of 1 ps, and assuming that the fraction *q_h_* of the hot electrons in the plasma is proportional to the plasma extension at the end of the laser pulse $\ell_{P}\left(t_{P}\right)$, this leads to the heuristic modification of the fraction of hot electrons as
(7)$$q_{h,mod}\;=\;q_{h}\cdot {\left(\frac{t_{p}}{1ps}\right)}^{3/4}.$$

With Equation (4), the modified number density of the hot electrons is finally found to be
(8)$$n_{h,mod}\;=\;Z_{i}\cdot n_{i}\cdot q_{h,mod}\;=\;Z_{i}\cdot n_{i}\cdot q_{h}\cdot {\left(\frac{t_{p}}{1ps}\right)}^{3/4}.$$

### 2.4. Dose Rate

In this section the dose rate $\dot{H}\left(d_{p}\right)$ inside an absorbing body, which results from an X-ray emission, is considered. *d_p_* is the penetration depth at which the dose rate is measured. The calculation for the dose rate, which was presented in [19], is now extended to account for the influence of the pulse duration on the number density of the hot electrons by replacing *n_h_* with *n_h,mod_* as defined in Equation (8), yielding
(9)$$\dot{H}\left(d_{P}\right)\;=\;\frac{f_{L}\cdot t_{P}\cdot V_{P}}{\rho_{ICRU}\cdot 2\pi \cdot d_{D}^{2}}\cdot \frac{32\pi }{3}\cdot \sqrt{\frac{2\pi }{3k_{B}\cdot T_{h}}}\cdot \frac{Z_{i}\cdot e^{6}\cdot n_{h,mod}^{2}}{m_{e}^{3/2}\cdot c^{3}}\cdot \int_{\omega \;=\;0}^{\omega =\;\infty }T_{A,F}\left(\omega \right)e^{-\frac{\hbar \cdot \omega }{k_{B}\cdot T_{h}}}\frac{1}{\ell_{a,ICRU}\left(\omega \right)}e^{-\frac{d_{p}}{\ell_{a,ICRU}\left(\omega \right)}}\cdot d\omega ,$$
where $\rho_{ICRU}$ = 1 g·cm^−3^ and $l_{a,ICRU}$ are the density and the optical penetration depth, respectively, of the International Commission on Radiation Unions (ICRU) sphere, which is used as reference for an absorbing body to calculate the dose rate. *d_D_* denotes the distance of the detector from the plasma, *f_L_* is the laser pulse frequency, and *T_A,F_* is the spectral transmission through air and all filtering materials between the plasma and the detector. It is noted that the unit for the dose rate $\dot{H}\left(d_{p}\right)$ as used in Equation (9) is Gy/h. The unit Sv/h of the dose rate, as used in radiation protection, includes weighting factors, which account for the damage that different types of radiation might cause to different organs. However, in the case of frequent X-ray radiation, the approximation 1 Sv = 1 Gy is valid [27] and is used below.

## 3. Experiments and Verification of the Model

### 3.1. Experimental Setup

To include possible laser system-specific effects on X-ray generation as was shown for the chirp of a laser pulse [28], three completely different laser systems were used for the experiments. The target material was stainless steel (1.4301) in all cases. The resulting $\dot{H}$(0.07) dose rates were measured with spectrally integrating OD-02 dosimeters (STEP Sensortechnik und Elektronik Pockau GmbH, Pockau-Lengefeld, Germany), which were covered with an 8 µm thick beryllium foil to block environmental light. The dosimeters are sensitive to photons in the range from about 1 keV to 25 MeV, depending on the operation mode [29].

Great care was taken to record the unaltered maximum dose rates. Since some material evaporates from the sample with every laser pulse, a small groove is formed during the process, which can lead to shielding of the emitted X-radiation by the walls of the groove. This shielding effect may cause a significant reduction in the measured dose rates. With all three laser systems, a square area was processed by line scanning. The scanning direction was varied in increments of 45° for each processed plane. The line overlap and the pulse overlap were kept constant at about 75% to avoid the creation of deep grooves. This procedure also reduces shielding due to remaining particles and is consistent with that proposed in [11]. Additionally, line scanning of the squared area was performed several times until the maximum measured dose rate was reached. This ensured that the maximum possible dose rate was measured. To ease comparison of the results, only this maximum dose rate was used, and it is presented below.

A Ti:Sapphire laser (Spitfire ACE, Newport Spectra Physics GmbH, Darmstadt, Germany) was used to investigate the dependence of the X-ray emission on the pulse duration at constant irradiance. The laser provided pulses with durations between 75 fs and 6 ps, with a maximum pulse energy of up to 7 mJ at the wavelength of 800 nm (FWHM = 30 nm) at the constant repetition rate of 1 kHz. A beam with a raw diameter of 10 mm was focused with an F-Theta lens (Sill Optics GmbH & Co. KG, Wendelstein, Germany) with a focal length of 100 mm to a diameter of 27 µm on the processed surface. The irradiance was kept constant at *I_0_* = 3.6 × 10^14^ W/cm^2^ by adapting the pulse energy, i.e., keeping the ratio of pulse energy and pulse duration constant. The OD-02 dosimeter was positioned at a distance of 25 cm from the processed area.

Experiments at constant pulse energy and constant average power were performed using a Yb:YAG laser system (Pharos, Light Conversion, Vilnius, Lithuania) at a wavelength of 1030 nm. The laser was operated with pulse durations between 250 fs and 5 ps, at a constant pulse repetition rate of 50 kHz, and at the maximum average power of 14 W, i.e., with a pulse energy of 280 µJ. A beam with a raw diameter of 5 mm was focused with an F-Theta lens with a focal length of 163 mm to a focal diameter of 52 µm on the processed surface, resulting in irradiance between 5.4 × 10^12^ W/cm^2^ and 1.1 × 10^14^ W/cm^2^, depending on the pulse duration used. The X-ray dose rates were measured at a distance of 20 cm. 

Experiments using both constant irradiance and constant average power were made with a second Yb:YAG laser system (TruMicro 2000 TRUMPF SE + Co. KG, Ditzingen, Germany) operating at a wavelength of 1030 nm. A beam with a diameter of 8 mm on the F-Theta lens was focused to a diameter of 22.4 µm on the workpiece. The pulse durations ranged from 300 fs to 20 ps. The irradiance was set to 1.6 × 10^13^ W/cm^2^ for the experiments using constant irradiance. The constant average power of 18.8 W at a repetition rate of 200 kHz, i.e., with a pulse energy of 94 µJ, resulted in irradiance between 2.4 × 10^12^ W/cm^2^ and 1.6 × 10^14^ W/cm^2^, depending on the pulse duration used. The X-ray dose rates were measured at a distance of 20 cm. 

All focused beam diameters were measured using the method of Liu [30].

### 3.2. Results

Figure 1 shows the $\;\dot{H}$(0.07) dose rate, normalized with the average laser power, measured using the Spitfire laser system (red squares) and the TruMicro laser system (blue triangles) as a function of the pulse duration.

The irradiance was kept constant at 3.5 × 10^14^ W/cm^2^ and 1.6 × 10^13^ W/cm^2^ by adapting the pulse energy. Normalizing the dose rate with the average laser power eases the comparison of data from different processing stations as the dose rate scales linearly with the average laser power [19]. The error bars include the estimated uncertainties of the pulse durations that were measured using an autocorrelator and those of the transmission through optical elements, as well as those of the dose rates as measured with the OD-02 dosimeter, including the uncertainty of focal diameter and shielding effects. It is seen that the measured dose rates increase by about three orders of magnitude when the pulse duration is increased by two orders of magnitude.

The solid lines are the dose rates calculated with the extended model as given in Equation (9) using the values of the experimental settings and calibration in [19]. It is found that the modified model is in excellent agreement with the experimental data despite the use of two completely different experimental setups with lasers operating at two different wavelengths.

A wide range of pulse durations and resulting irradiances were available with the two Yb:YAG laser systems. For the results shown in Figure 2, both the repetition rate and the pulse energy, and, hence, the average power, were kept constant. The average power was 14 W for the Pharos laser-system (red squares) and 18.8 W for the TruMicro system (blue triangles). Figure 2 shows the measured $\dot{H}$(0.07) dose rates divided by the average laser power as a function of the pulse duration. The focal diameters of 52 µm (Pharos) and 22.4 µm (TruMicro) were kept constant. The resulting change in the irradiance in Figure 2b was therefore solely due to the changed pulse duration.

The measured dose rate decreases by about three orders of magnitude when the pulse duration is increased from 250 fs to 5 ps or from 300 fs to 8 ps (**a**), or, correspondingly, when the irradiance is decreased from 1.6 × 10^14^ W/cm^2^ to 5.4 × 10^12^ W/cm^2^ (**b**).

It is again found that the model given in Equation (9) is in excellent agreement with the experimental data for constant average power and two completely different experimental setups.

The fact that at a given pulse duration, or at a given irradiance, the smaller diameter of the laser beam of 22.4 µm (blue) leads to larger dose rates than those obtained with the larger beam diameter of 52 µm (red) is no contradiction to Equation (9), according to which larger plasma volumes *V_P_* emit more X-radiation. The corresponding results shown in Figure 2 can be explained by the experimental preconditions:As shown in Figure 2a, at a given pulse duration, a smaller beam diameter (blue values) results in a higher irradiance. The increase in the X-ray emission with increasing irradiance is very strong, overriding the effect of the larger plasma volume.In Figure 2b, at a given irradiance, a smaller beam diameter (blue values) means that a longer pulse duration was applied. Figure 1 shows the increase in the X-ray emission with a pulse duration that is quite strong. Since the increase in the X-ray emission with increasing plasma volume *V_P_* is slightly less pronounced than the increase in the X-ray emission with increasing pulse duration, the values obtained with the larger beam diameter and shorter pulses (red) are slightly lower.

## 4. Conclusions

The dependence of the $\dot{H}$(0.07) dose rate on the duration of laser pulses on solid samples was investigated experimentally and analytically. The analytical model presented in [19] was extended to account for the influence of pulse duration on the fraction of hot electrons present in the generated plasma. The extended model was verified by experiments involving three completely different laser systems. The experiments were performed at wavelengths of 1030 nm and 800 nm, with pulse repetition rates of 200 kHz, 50 kHz, and 1 kHz. The pulse duration was varied from 100 fs to 8 ps, resulting in a range of irradiance from 5.4 × 10^12^ W/cm^2^ up to 3.6 × 10^14^ W/cm^2^.

The significant decrease in the $\dot{H}$(0.07) dose rates with a decreasing pulse duration predicted by the model was confirmed with very good agreement in the experiments. This result agrees with the assumption that the interaction of the laser pulse with plasma electrons that are already present during the duration of the pulse strongly affects the generation of hot electrons and, with it, the amount of X-ray emission.

The findings might also explain the increased dose rates that were measured during processing using bursts of laser pulses (200 fs pulses) with intraburst pulse repetition rates of 66.7 MHz [15]. The first pulse in such a burst creates a plasma that is still partly present when the subsequent pulse arrives, leading to enhanced creation of hot electrons. This agrees with the approximate threefold increase in the X-ray emission caused by small pre-pulses as reported in [31]. However, within the experimental uncertainty, the dose rates measured with burst of fs pulses [15] are the same as those that are measured when the total energy of the burst is applied in a single pulse with a duration of 1 ps.

## Figures and Tables

**Figure 1 materials-15-02257-f001:**
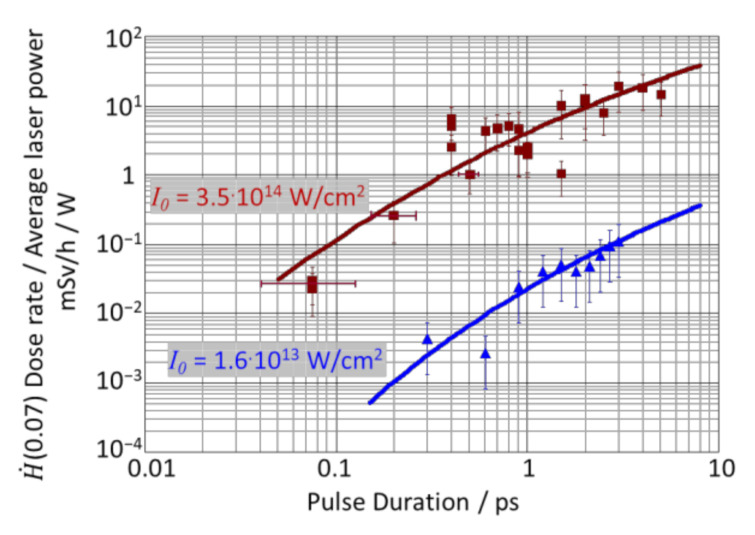
Measured $\dot{H}$(0.07) dose rates divided by the incident average laser power as a function of the pulse duration for a constant irradiance of 3.6 × 10^14^ W/cm^2^ (squares) and 1.6 × 10^13^ W/cm^2^ (triangles). The dose rates were measured with an OD-02 dosimeter at a distance of 25 cm (red squares) and 20 cm (blue triangles). The irradiance was kept constant by adapting the pulse energy. The solid lines were calculated with the extended model given in Equation (9) using the values of the respective experimental settings.

**Figure 2 materials-15-02257-f002:**
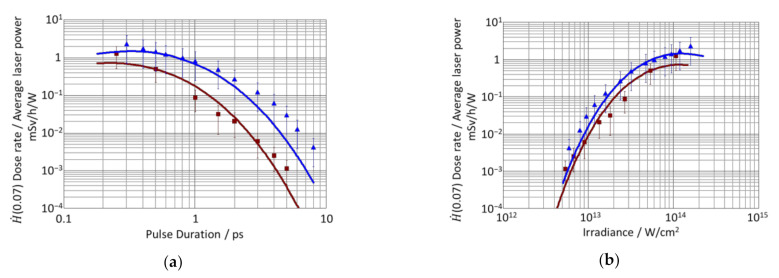
Measured $\dot{H}$(0.07) dose rates divided by the average laser power as a function of the pulse duration (**a**) and as a function of the irradiance (**b**) for constant average laser powers of 14 W (red squares) and 18.8 W (blue triangles) at a distance of 20 cm from the plasma. The solid lines were calculated with the extended model given in Equation (9) using the values of the respective experimental settings.

## Data Availability

Data sharing is not applicable to this article.

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
