# Peer review of "Influence of Pulse Duration on X-ray Emission during Industrial Ultrafast Laser Processing"

_materials, 2022, doi:10.3390/ma15062257_

Round 1
Reviewer 1 Report
This is a nicely written and useful paper and I recommend it for publication after minor revision. I only have few comments:
1) The effect of density is not discussed. It may not be relevant for the conditions studied here but it is of general interest: The plasma frequency of the quasi-free electrons shift by (OM) 0.5 eV over 50 GPa for group VIIIb elements. It would be useful to take this effect into account.
2) an initial density of the inertially confined plasma of 1 g/cm3 is a gross under-estimation. Could you add calculations with more realistic densities?
3) btw. the term 'ICDU' is not defined as far as i can tell. This needs to be added.
4) Is the release of electrons into the plasma state element specific - different for Fe,Cr,Ni in the alloy? Can you specify the emitted X-rays by characteristic lines?
5) lines 183, 231, 235, and 237: references missing!
Reviewer 2 Report
The paper is fine and interesting for laser applications.
The study of laser duration on the emission of X-rays is interesting and also the modified model including laser pulse length.
Maybe the systematic errors could be discussed in more details but overall the paper could be published in the current form.
There are some points to be improved (see file).

Reviewer 3 Report
It is important study of X-ray generation at the irradiances used often at laser machining. Several not clear points:
• pulse duration can is setup by changing chirp of ultra-short pulse. the same duration can be achieved by positive or negative chirp, however, X-ray generation is markedly different (Optics Express Vol. 16, Issue 17, pp. 12650-12657 (2008)
•https://doi.org/10.1364/OE.16.012650). What was X-ray emission for positive and negative chirp?
• has X-ray spectrum measured and how hard-X rays contribution was avoided. Hard X rays had to be present at the used exposure as demonstrated using femtosecond irradiation of PbO glass.The model is presented with a fit parameter for fraction of hot electrons. It would be useful to present the efficiency of X-ray generation: Number of Xray photons vs number of laser photons.
• The H(0.07) might be understood as dp = 0.07 since equation has H(dp). this is not clear
• there are number of types for missing citations
